# Activation of Protein Kinase A (PKA) signaling mitigates congenital hyperinsulinism associated hypoglycemia in the *Sur1<sup>-/-</sup>* mouse model

**Mangala M. Soundarapandian[1]\*, Christine A. Juliana[2], Jinghua Chai[2], Patrick A. Haslett[1], Kevin Fitzgerald[1], Diva D. De León[2,3]\***

1 Alnylam Pharmaceuticals, Cambridge, Massachusetts, United States of America, 2 Division of Endocrinology and Diabetes, The Children's Hospital of Philadelphia, Philadelphia, Pennsylvania, United States of America, 3 Department of Pediatrics, Perelman School of Medicine at the University of Pennsylvania, Philadelphia, Pennsylvania, United States of America

\* msoundar@alnylam.com (MMS); deleon@email.chop.edu (DDDL)

**Data Availability Statement:** All relevant data are within the manuscript and its Supporting Information files.

## Abstract

There is a significant unmet need for a safe and effective therapy for the treatment of children with congenital hyperinsulinism. We hypothesized that amplification of the glucagon signaling pathway could ameliorate hyperinsulinism associated hypoglycemia. In order to test this we evaluated the effects of loss of Prkar1a, a negative regulator of Protein Kinase A in the context of hyperinsulinemic conditions. With reduction of Prkar1a expression, we observed a significant upregulation of hepatic gluconeogenic genes. In wild type mice receiving a continuous infusion of insulin by mini-osmotic pump, we observed a 2-fold increase in the level of circulating ketones and a more than 40-fold increase in Kiss1 expression with reduction of Prkar1a. Loss of Prkar1a in the *Sur1<sup>-/-</sup>* mouse model of $K_{ATP}$ hyperinsulinism significantly attenuated fasting induced hypoglycemia, decreased the insulin/glucose ratio, and also increased the hepatic expression of Kiss1 by more than 10-fold. Together these data demonstrate that amplification of the hepatic glucagon signaling pathway is able to rescue hypoglycemia caused by hyperinsulinism.

## Introduction

Congenital Hyperinsulinism (HI) is a genetic disorder of the pancreatic β-cells that causes dysregulated insulin secretion and persistent hypoglycemia. There are at least nine different genetic subtypes of hyperinsulinism, but the most common and severe form is caused by inactivating mutations in *ABCC8* or *KCNJ11*, the genes encoding the two component of the β-cell $K_{ATP}$ channel [1]. HI is the most common cause of persistent hypoglycemia in neonates, infants and children and is associated with high risk for serious complications (seizures, intellectual deficiencies, brain damage, and coma) with the rate of neurodevelopmental deficits in these patients as high as 48% [2]. Only about 40% of patients respond to the limited number of

**Funding:** This study was supported by Alnylam Pharmaceuticals. https://www.alnylam.com MMS, PAH and KF work for Alnylam Pharamaceuticals; MMS, PAH and KF hold shares in Alnylam Pharmaceuticals stock; DDDL received funding from Alnylam Pharmaceuticals to conduct this study. MMS, PAH, KF and DDDL collaborated on the design, data collection, analysis, decision to publish and preparation of the manuscript. JC performed experiments and CAJ analyzed data and wrote the manuscript.

**Competing interests:** MMS, PAH and KF work for Alnylam Pharamaceuticals; MMS, PAH and KF hold shares in Alnylam Pharmaceuticals stock; DDDL received funding from Alnylam Pharmaceuticals to conduct this study. We confirm that this does not alter our adherence to the PLOS ONE policies on sharing data and materials and have updated our Competing Interests statement.

existing therapies and this is not without significant limitations and side effects [3–5]. Thus, there is a serious and unmet need for development of safe and effective therapies for treatment of HI.

The liver offers several possible avenues for therapeutic intervention for HI because of its central role in systemic glucose homeostasis through regulation of glycogen storage, gluconeogenesis, and suppression of insulin secretion through production of the hepatokine, kisspeptin1 (KISS1). Protein Kinase A (PKA) is a serine/threonine kinase that is inactive while bound to a dimer composed of regulatory subunits (e.g. Prkar1a) [6]. When plasma glucose concentration falls below 65–70 mg/dL, glucagon is secreted from pancreatic alpha cells and binds to its receptor on hepatocytes, which leads to binding of cAMP to the regulatory subunits, conformational changes, and the release of active PKA [6, 7]. Active PKA promotes glycogenolysis through glycogen phosphorylase kinase (PhK) and gluconeogenesis through the increased expression of key genes [phosphoenolpyruvate carboxykinase (PEPCK), glucose 6-phosphatase (G6Pase), peroxisome proliferator-activated receptor gamma coactivator 1-alpha (Ppargc1a, PGC1α)] [7–13]. Stimulation of cAMP-PKA-CREB signaling by glucagon upregulates KISS1. Kiss1 is a secreted hepatokine that signals via the kisspeptin receptor on pancreatic β-cells to decrease insulin secretion [14]. Thus, due to the central role played by PKA in glucagon signaling, its disinhibition by specific depletion of Prkar1a in the liver leads to a significant abrogation of glucose stimulated insulin secretion and increased plasma glucose [14].

Given the ability of disinhibited PKA to increase plasma glucose levels, we wanted to evaluate the effect of Prkar1a reduction in the liver on hypoglycemia caused by hyperinsulinemic conditions. We found that siRNA mediated loss of Prkar1a increased ketone levels and induced KISS1 expression in the context of exogenous insulin treatment. Strikingly, in the $Sur1^{-/-}$ mouse model (lacking the Sulfonylurea receptor1 subunit of the $K_{ATP}$ channels and thus a model of $K_{ATP}$ hyperinsulinism), we found that reduction of Prkar1a resulted in a significant decrease in plasma insulin and an attenuation of fasting hypoglycemia. These findings identify a new critical nexus for development of therapies for treatment of hypoglycemia in children of HI.

## Materials and methods

### Animal studies

Wildtype rodent studies were conducted at Alnylam Pharmaceuticals and $Sur1^{-/-}$ mice studies were conducted at the Children's Hospital of Philadelphia and approved by the Institutional Animal Care and Use Committee (IACUC) of the respective institutions. Method of euthanasia: Inhalation of carbon dioxide (CO2) followed by cervical dislocation.

The generation and genotyping of $Sur1^{-/-}$ mice were previously described [15]. $Sur1^{-/-}$ mice are bred and maintained in our mouse colony on a C57Bl/6 genetic background for experiments. 7 male mice, eight to ten weeks old, $Sur1^{-/-}$ mice were used in each group in all experiments. Mice were maintained on a 12:12-h light-dark cycle and were fed a standard rodent chow diet with free access to food and water. Animal welfare checks were completed once per day. No adverse effects to these experiments were observed or required analgesia.

Wildtype (WT) C57BL/6J female mice were acquired from Jackson laboratories, (Bar Harbor, ME) at 8–10 weeks of age. 6–8 WT mice were used per group in experiments. The each dot in the scatter plots represent one animal each. Mice were allowed to acclimate to a 12:12-h light-dark cycle, housing humidity and temperature for at least 72 hours prior to initiation of the study. Mice were group-housed and maintained on a standard rodent diet (LabDiet, Picolab rodent diet 5053). All animals were provided free access to drinking water. Animal welfare

checks at Alnylam are conducted every 24hrs. We did not observe any adverse events at the dose of siRNA used in these studies.

## Glycogen staining

For glycogen staining the mice were subcutaneously injected with vehicle control (1X PBS) or indicated doses of Prkar1a siRNA. Glycogen was detected in liver sections following a standardized periodic acid Schiff (PAS) staining technique. Briefly, livers fixed in 10% neutral buffered formalin were embedded in paraffin blocks. 4-micron sections were collected on glass slides, de-paraffinized, incubated with 0.5% periodic acid for 7 min, rinsed in water, and placed in Schiff's reagent for 15 min. Finally, slides were washed with water and nuclei were stained with Modified Mayer's Hematoxylin. 1% Diastase was used to verify that staining was specific for glycogen. All reagents were obtained from Rowley Biochemical.

## Evaluation of glucose homeostasis

For $Sur1^{-/-}$ mice random fed plasma glucose and morning fasting (16 hrs) plasma glucose, plasma betahydroxybutyrate and plasma insulin concentrations were measured at baseline and at days 4 and weekly for 3 weeks after treatment. Glucose tolerance testing was carried after a 16 hour fast by administering 2g/kg of dextrose intraperitoneally. Plasma glucose and betahydroxybutyrate concentrations were measured using a hand-held glucose meter (NOVA, Nova Biomedical). Plasma insulin was measured by ELISA (ALPCO; catalogue #80-INSMS-E01).

For wildtype mice, fasting plasma glucose and betahydroxybutyrate were assessed after a 5hr morning fast using blood from a tail nick using handheld glucose meter (ACCU-CHEK Aviva, Roche) or ketone meter (Precision Xtra, Abbott). Fed glucose was measured at the end of the dark cycle. Pyruvate tolerance test was carried out after a 14hr overnight fast by administering 1.5 g/kg sodium pyruvate intraperitoneally.

## Osmotic pump implantation

Alzet Micro-osmotic pumps, model 1002 with pumping rate 0.25μl/hr (DURECT Corporation) were filled with Humulin (Eli Lilly) diluted in 1X sterile PBS to allow Insulin release of 0.2 or 0.3U/day. Pumps were implanted subcutaneously under isoflurane anesthesia. The mice were allowed to recover and their plasma glucose was monitored using glucose meter (ACCU-CHEK Aviva, Roche). Plasma insulin levels was measured by ELISA (Crystal Chem, Ultrasensitive mouse insulin ELISA kit, Catalogue #90080, Lot # 16SEUMI411 that detects both human and mouse insulin)

## siRNA injection studies

For Prkar1a knockdown studies the mice were subcutaneously injected with vehicle control (1X PBS) or 1 mg/kg Prkar1a siRNA every 2 weeks unless otherwise indicated. The endpoints were assessed from serum or liver tissue 28 days post dosing unless otherwise indicated.

The siRNA targeting Prkar1a was designed, synthesized, and liver targeted by Alnylam Pharmaceuticals, as previously described [16, 17]. The siRNA was designed to target mouse Prkar1a mRNA, NM_021880.

AD-76409 targets position 865–885, 5'-GAUGUAUGAAGAAUUCCUUAGUA-3'
AD-76410 targets position 873–893, 5'-AAGAAUUCCUUAGUAAAGUGUCU-3'
AD-76411 targets position 1394–1414, 5'-AAAAGUUGCUUUAUUGCACCAUU-3'

## RNA isolation and qRT-PCR

Total RNA was isolated from liver tissue using the miRNeasy kit (Qiagen) following manufacturer's protocols. 1ug of RNA from each sample was reverse transcribed using the High capacity Reverse transcription kit (Invitrogen). Quantitative real time PCR was performed on the cDNA using Roche light cycler and the Lightcycler 480 master mix (Roche). All experimental samples were analyzed and normalized with the expression level of a reference gene [calculated by second-derivative maximum by applying the 2−(ΔΔCt) method]. The following Taqman assays (Invitrogen) were used: Prkar1a (Mm00660315_m1), G6PC (Mm00839363_m1), PEPCK (Mm01247058_m1), Ppargc1a (Mm01208835_m1), Kiss1 (Mm03058560_m1), GCK (Mm00439129_m1) and Gapdh (4352339E)

## Western blot analysis

Livers were homogenized in RIPA buffer along with protease inhibitors. Total cell lysates denatured by boiling in 2x Laemmli buffer were subjected to SDS-PAGE and transferred to nitrocellulose membranes. The blots were hybridized to specific antibodies overnight at 4˚C and the bands were detected using fluorescence imaging using the Licor system. The following antibodies were used: Prkar1a (BD biosciences, Catalog #610609, 1:500), PKAC (BD Biosciences, Catalog #610980, 1:2000), β-actin (Abcam, Catalog #ab8227, 1:2000), PKA substrate (Cell Signaling, Catalog #9624, 1:1000), Fluorescence conjugated secondary antibodies (Licor, Goat anti-rabbit, Catalog #926–32211 and Donkey anti-mouse, Catalog #926–680721:5000).

## Statistics

Statistical analyses were performed on GraphPad Prism 6 software. All results are presented as mean ± standard error of the mean (SEM). The level of significance was set at $P < 0.05$. For multiple measurements data were analyzed using 2-way ANOVA Repeated Measures, Tukey's multiple comparison test. Single time end points data were analyzed using one-way ANOVA or Student's t-test.

# Results

## Loss of Prkar1a activates PKA and downstream liver gluconeogenesis

In order to achieve reduction of Pkar1a in the liver, mice were subcutaneously injected with a liver-targeted siRNA directed against *Prkar1a* or PBS control. Liver extracts harvested at 10 or 28 days post injection with either 0, 0.5, 1, 3, or 5 mg/kg doses of siRNA revealed a dose dependent suppression of Prkar1a mRNA. The lowest dose of siRNA (0.5 mg/kg) demonstrated a ~60% or ~75% reduction of Prkar1a mRNA expression at 10 and 28 days post initial injection, respectively (Fig 1A). A ~90% reduction of Prkar1a mRNA is achieved by 3 or 5 mg/kg doses at both time points (Fig 1A). Subcutaneous injection of siRNA (1 mg/Kg, bi weekly) directed against Prkar1a also effectively reduced PRKAR1A protein while not having a significant effect on catalytic PKA (PKAc) protein levels (Fig 1B).

A previous study found increased glycogenolysis and gluconeogenesis in hepatic cells from mice expressing constitutively active PKA [18]. Here we demonstrate that direct loss of the PKA regulatory subunit, Prkar1a, increased PKA activity as evidenced by an increase in phosphorylation of PKA substrates (S1 Fig) and upregulation of expression of downstream targets important for gluconeogenesis: glucose-6-phosphatase (G6Pase), phosphoenolpyruvate carboxykinase (PEPCK), and PPARγ coactivator-1α (PGC-1α) (Fig 1C). Functionally, the loss of Prkar1a resulted in an increase in glycogenolysis and gluconeogenesis as observed by a significant reduction in liver glycogen (Fig 1D) and though not statistically significant, a trend of

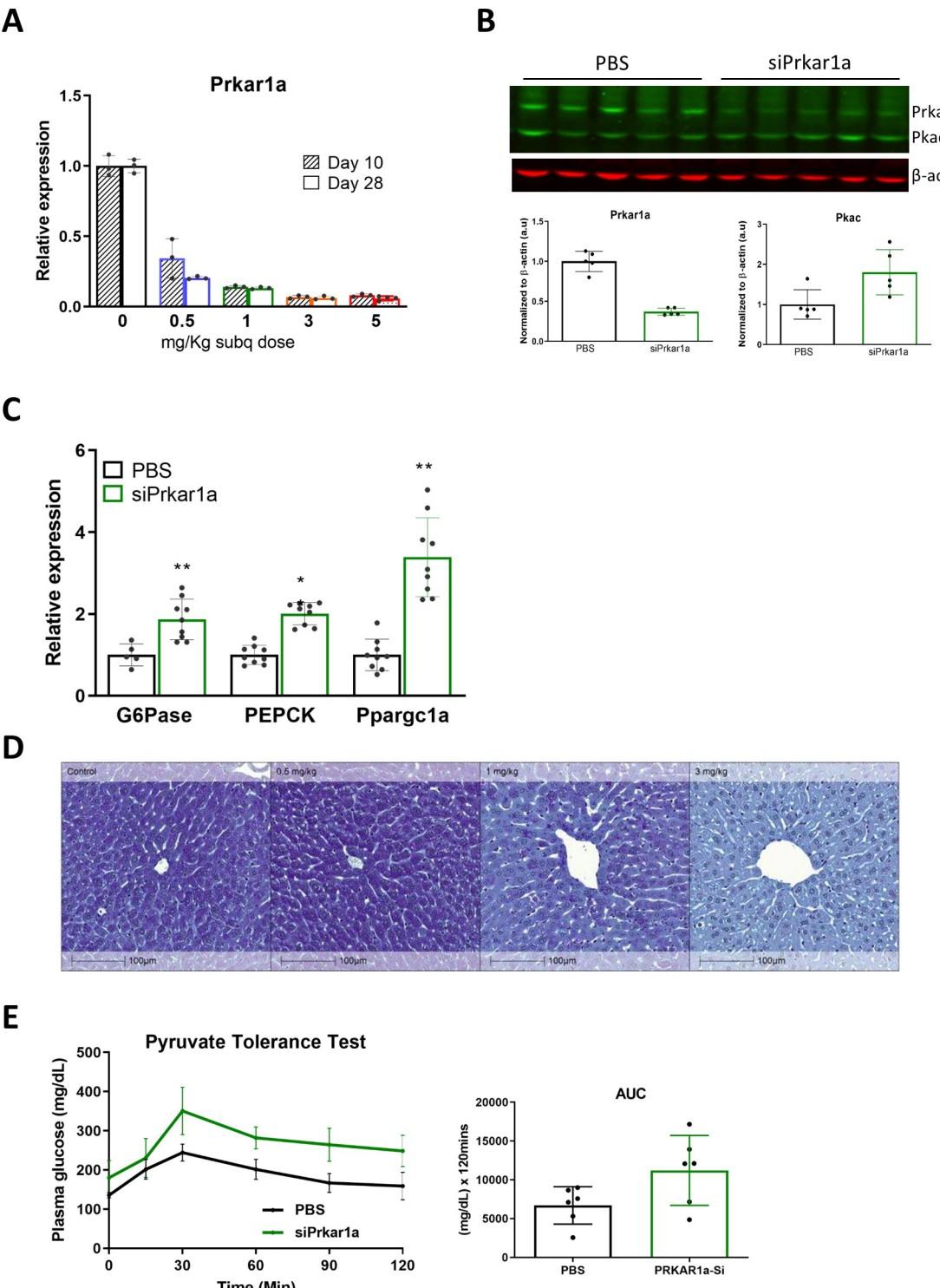

**Fig 1. siRNA mediated reduction of Prkar1a activates PKA and liver gluconeogenesis.** WT mice were injected subcutaneously with siPrkar1a (AD-76410) at the denoted mg/kg dose. Liver extracts were collected from siRNA injected mice or PBS controls at either 10 or 28 days post-injection for (A) qRT-PCR analysis of Prkar1a mRNA expression, or (B) protein for western blot analysis of PRKAR1A and catalytic PKA expression with calculated relative densities normalized to β-actin from liver extracts from WT mice 28 days post-injection. (C) qPCR analysis of mRNA expression of gluconeogenesis targets G6Pase, PEPCK, and Ppargc1a of RNA extracted from liver extracts of bi-weekly siPrkar1a (1 mg/kg) injected WT mice compared to vehicle controls. (D) Glycogen staining of liver tissue in WT mice injected with the denoted dose of siPrkar1a. (E) Pyruvate tolerance test in WT mice administered

21 days after injection with siPrkar1a or PBS control after a 14 hour overnight fast, with calculated area under the curve (AUC). (n = 6 mice/group) Data represent mean +/- SEM. *, p ≤ 0.05; **, p ≤ 0.01 compared to PBS control.

increased conversion of the gluconeogenic precursor pyruvate to glucose in a pyruvate tolerance test (Fig 1E) was observed in mice injected with siPrkar1a. These results demonstrate the essential role of Prkar1a regulation of PKA in glycogenolysis and gluconeogenesis, as well as show that reduction of Prkar1a can mimic the effects of glucagon signaling through the increase of liver glucose output.

## Reduction of Prkar1a leads to hyperglycemia in mice

As siRNA mediated loss of Prkar1a caused increased glycogenolysis, we assessed the effect on plasma glucose concentration in mice injected with siPrkar1a. Three different siRNAs directed at Prkar1a resulted in a decrease of Prkar1a transcript with different potencies. AD-76409 and AD-76410 achieved ~90% reduction of Prkar1a mRNA, while AD-76411 only resulted in ~30% loss (Fig 2A). Significant hyperglycemia was observed in both fasting and fed states compared to control starting at 10 days and enduring until 28 days post-injection in wildtype mice treated with siPrkar1a sequences in a manner correlating with the potency of Prkar1a reduction (Fig 2B and 2C). Injection of AD-76410 at different doses (0.5, 1, 3, or 5 mg/Kg) over 28 days at 2 week intervals demonstrated a dose dependent increase in plasma glucose concentration compared to controls. All siPrkar1a doses increased plasma glucose concentration in a dose dependent manner, with the highest doses (3 and 5 mg/Kg) resulting in plasma glucose concentrations of ~400 mg/dL and the lowest dose (0.5 mg/Kg) with plasma glucose concentrations of ~300 mg/dL by day 21 post-injection (Fig 2D) compared to PBS injected control mice in which plasma glucose concentration never exceeded 200 mg/dL. In mice receiving a single dose of siRNA, only the highest doses (5, 3, and 1 mg/Kg) resulted in a significant increase in plasma glucose for the entire 28 days of assessment (S2A Fig). The most potent siRNAs against Prkar1a (AD-76409 and AD-76410) led to a significant upregulation of gluconeogenesis genes PEPCK and PGC-1α (Fig 2E). AD-76409 was also able to significantly increase G6Pase expression. Additionally, ketone levels (β-hydroxybutyrate) trended higher, although not statistically significant in mice treated with the highest siRNA doses (S2B Fig). There was no significant change in weight in these mice compared to controls (S2C Fig). The application of three independent siRNAs directed against Prkar1a and the correlating effects on plasma glucose based dose dependence and potency of reduction indicate the siRNA mediated knockdown of Prkar1a requires reduction of greater than 30% to increase plasma glucose levels.

## Prkar1a loss increases plasma ketones during hyperinsulinemic conditions and induces *Kiss1* expression

The ability of Prkar1a reduction to significantly increase plasma glucose made it an interesting target for treatment of hypoglycemic conditions. To determine the effectiveness of loss of Prkar1a on hyperinsulinemic hypoglycemia, mice were implanted with an osmotic pump delivering either vehicle, 0.2U or 0.3U/day of insulin 16 days after they were injected with siPrkar1a or control. As expected, insulin administration lowered blood glucose levels (solid lines). In mice pre-treated with Prkar1a siRNA, the baseline plasma glucose was higher (similar to Fig 2B) and we observed a dose dependent attenuation but not a reversal of insulin induced hypoglycemic effects (dotted lines, (Fig 3A). Loss of Prkar1a resulted in suppression of endogenous insulin levels, but as expected, there was no effect on exogenously administered insulin levels (Fig 3B). Interestingly, treatment with siPrkar1a did result in a significant increase in

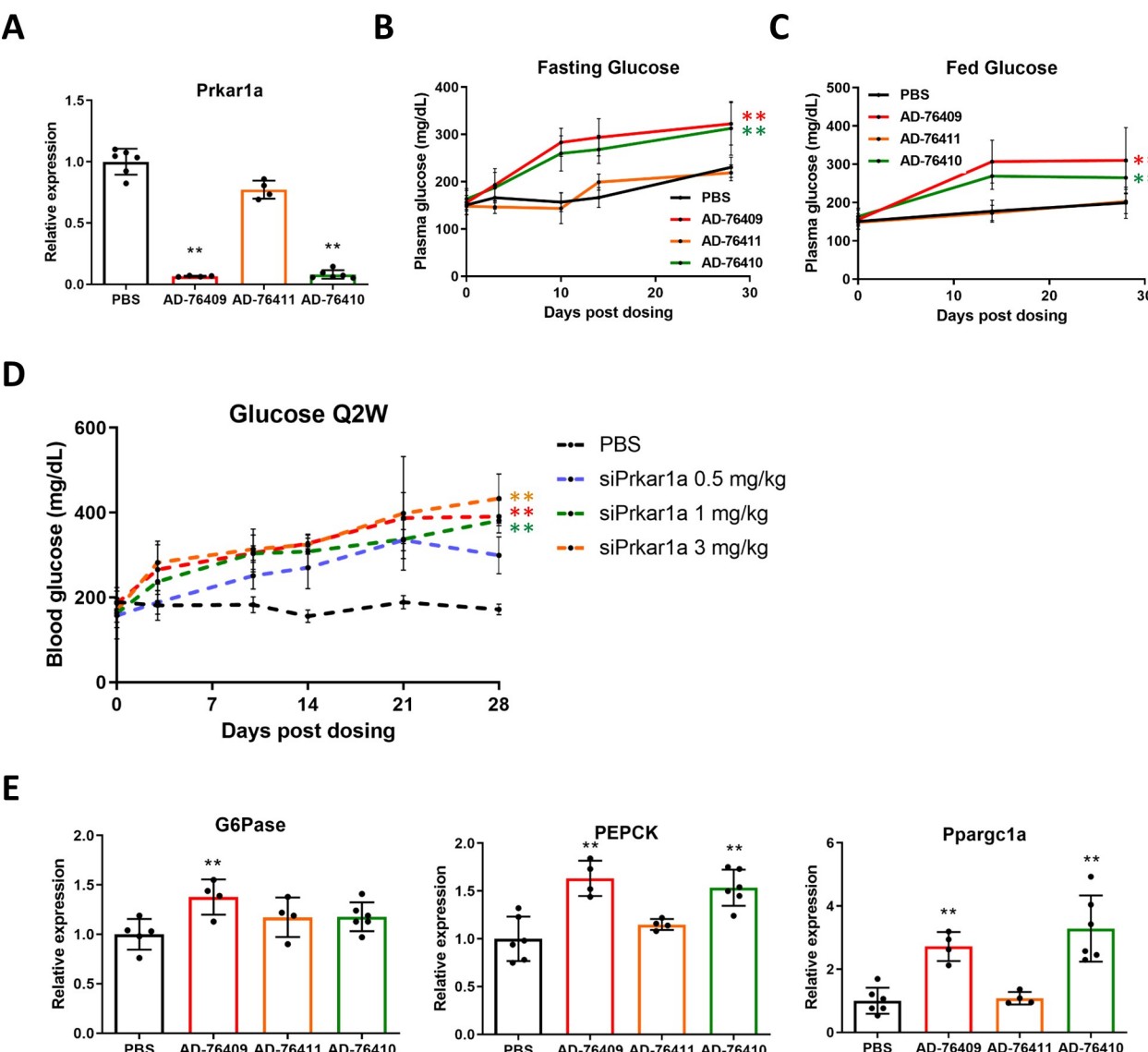

**Fig 2. Loss of Prkar1a results in hyperglycemia.** WT mice were injected subcutaneously with siRNA (1 mg/kg) directed against Prkar1a (AD-76409, AD-76411, AD-76410) with varying potencies and compared to PBS control. (A) Liver extracts were collected from siRNA injected mice or PBS controls for qRT-PCR analysis of Prkar1a mRNA expression. (B) Fasting plasma glucose levels were assessed after a 16 hour overnight fast at 0, 3, 10, 14, and 28 days post dosing with denoted siPrkar1a siRNAs. (C) Fed plasma glucose levels were assessed with ad libitum feeding at 0, 14, and 28 days post dosing with denoted siPrkar1a siRNAs. (D) Mice were injected with siRNA (AD-76410, 1 mg/kg) directed against Prkar1a every 2 weeks (Q2W) at the denoted mg/kg dose and plasma glucose levels were assessed at 0, 3, 10, 14, 21, and 28 days post dosing. (E) Liver extracts were collected from siRNA (AD-76410, 1 mg/kg) injected mice or PBS controls for RNA and qPCR analysis of mRNA expression of gluconeogenesis targets G6Pase, PEPCK, and Ppargc1a. (n = 6 mice/group) Data represent mean +/- SEM. *, $p \leq 0.05$; **, $p \leq 0.01$ compared to PBS control.

ketones only during insulin-induced hypoglycemic conditions, indicating reversal of the insulin-suppressive effect on ketogenesis (Fig 3C).

As seen in our earlier experiments, Prkar1a reduction led to a significant increase in gluconeogenic gene expression including G6pase, PEPCK, and Ppargc1a (Fig 3D). Hypoglycemia will stimulate the release of glucagon to activate hepatic pathways to restore normoglycemia, but most likely no additive effect is seen since maximal activation of these pathways is achieved by either. A previous study identified the hepatic production of the neuro-peptide kisspeptin1 (KISS1) by glucagon stimulation that resulted in the suppression of glucose stimulated insulin

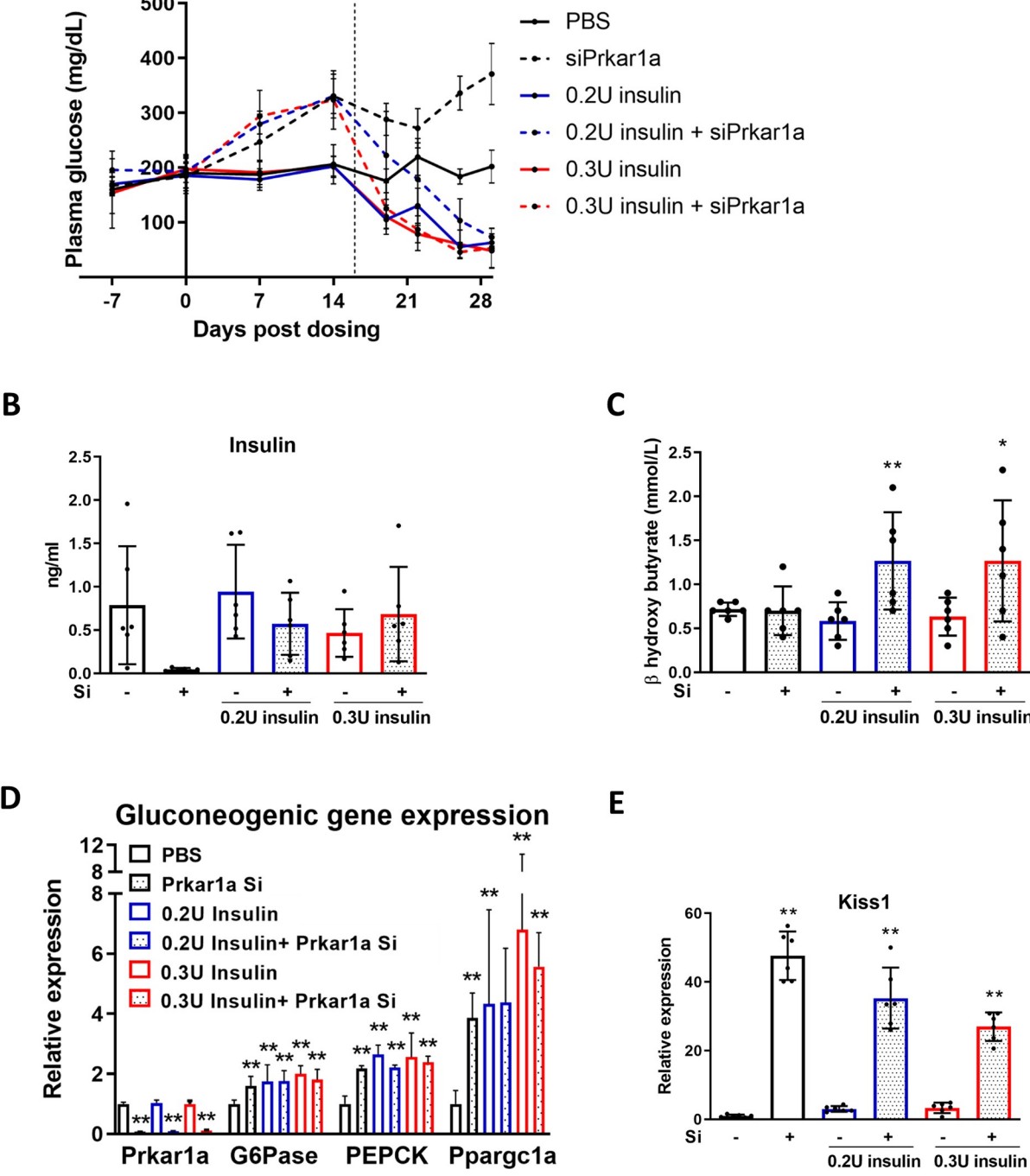

**Fig 3. Loss of Prkar1a increases blood ketones during hyperinsulinemic conditions.** WT mice were injected with siPrkar1a (AD-76410, 1 mg/kg) and a subcutaneous osmotic pump delivering either vehicle (1X PBS), 0.2U, or 0.3 U insulin/day was implanted 16 days (dotted line) post siRNA injection. (A) Plasma glucose levels were assessed 0, 7, 14, 19, 22, 26, and 29 days post dosing with siPrkar1a (AD-76410, 1 mg/kg) after a 5-hour morning fast. (n = 6 mice/group). (B) Plasma insulin levels were assessed at 13 days post osmotic pump implantation after a 5-hour morning fast. (C) Plasma β-hydroxybutyrate levels were assessed at 13 days post pump implantation after a 5-hour morning fast. (D) Liver extracts were collected from siRNA injected mice (AD-76410, 1 mg/kg) or PBS controls with osmotic pump implantation for RNA and qPCR analysis of mRNA expression of Prkar1a and gluconeogenesis targets G6Pase, PEPCK, and Ppargc1a. (E) Liver extracts were collected from siRNA injected (AD-76410, 1 mg/kg) mice or PBS controls with osmotic pump implantation for RNA and assessed by qPCR analysis for Kiss1 mRNA expression level. (n = 6 mice/group) Data represent mean +/- SEM. Statistics was calculated using Tukey's multiple comparisons test comparing PBS controls to siPrkar1a of same insulin dose. *, $p \leq 0.05$; **, $p \leq 0.01$.

secretion in β-cells [14]. Intriguingly, we found a strong induction by more than 40 fold of Kiss1 transcript with loss of Prkar1a (Fig 3E). The ineffectiveness of siPrkar1a to increase plasma glucose in this hypoglycemic model may be due to the use of an exogenous insulin source which is by definition not subject to regulation. We therefore proceeded to evaluate Prkar1a knockdown in the *Sur1*[-/-] mouse, an endogenous hyperinsulinemic model.

## Hypoglycemia is attenuated by loss of Prkar1a expression in the *Sur1*[-/-] mouse model of hyperinsulinism

*Sur1*[-/-] mice were injected with PBS or siPrkar1a and glucose homeostasis was evaluated in the course of 3 weeks. Fasting plasma glucose was significantly higher in siPrkar1a-treated compared to PBS-treated *Sur1*[-/-] mice 21 days after injection (Fig 4A). Fed plasma glucose was not significantly different between the two groups (Fig 4B). Absolute insulin concentration was not significantly different in the siPrkar1a treated *Sur1*[-/-] mice (S3A Fig), however, the insulin/glucose ratio was significantly decreased in the siPrka1a-treated compared to PBS-treated *Sur1*[-/-] mice (Fig 4C). Fasting ketones were not significantly different in siPrkar1a-treated compared to compared to PBS-treated *Sur1*[-/-] mice (S3B Fig). In response to a glucose tolerance test (GTT) siPrkar1a-treated *Sur1*[-/-] mice exhibited a significant increase of plasma glucose compared to controls as assessed by area under the curve (AUC) (Fig 4D). Insulin levels were significantly lower during the GTT in siPrkar1a-treated *Sur1*[-/-] mice, as determined by calculation of area under the curve (AUC) (Fig 4E). Importantly, reduction of Prkar1a led to a significant increase in the transcript levels of PKA targets (PEPCK and Ppargc1a) and pancreatic β-cell signaling hepatokine kisspeptin by more than 10 fold in the liver of *Sur1*[-/-] mice (Fig 4F and 4G). siPrkar1a was not able to overcome the suppression of G6pase in the setting of endogenous hyperinsulinemic condition. However, the ability of Prkar1a reduction to resolve fasting hypoglycemia in the *Sur1*[-/-] hyperinsulinism model exposes new possible therapeutic targets in the liver, PKA and glucagon signaling pathways in the treatment of hyperinsulinemic hypoglycemia.

## Discussion

Glucagon is used acutely to treat insulin-induced hypoglycemia, but its chemical and physical instability in solution has been a serious limitation for its use for extended periods of time in hormone pumps [19, 20]. Glucagon signaling increases plasma glucose concentration by activating protein kinase A (PKA) signaling that initiates the breakdown of liver glycogen stores and also through hepatic production of Kiss1, which directly suppresses insulin secretion in the β-cell. The present findings using a combination of mouse model of HI (*Sur1*[-/-]) and exogenous insulin administration demonstrate the importance of hepatic signaling in glucose homeostasis and the real potential of exploiting glucagon signaling as an untapped resource for treatment of HI associated hypoglycemia. Evaluation of mice in the context of hyperinsulinemic conditions by use of a mini-osmotic pump to administer exogenous insulin, demonstrated an almost doubling of circulating ketones (Fig 3). In normal fasting individuals, falling glucose concentrations stimulates ketogenesis as an alternative fuel source and this response is particularly important to prevent brain damage during hypoglycemia [21]. However, high levels of circulating insulin, such as that found in HI patients, inhibits ketogenesis and thereby removes this neuro-protective response to hypoglycemia [21–23]. Our results demonstrate that the ketogenic response can in fact be restored in hyperinsulinemic conditions by amplifying the glucagon signaling pathway sufficiently to surpass the inhibitory effects of insulin on ketogenesis.

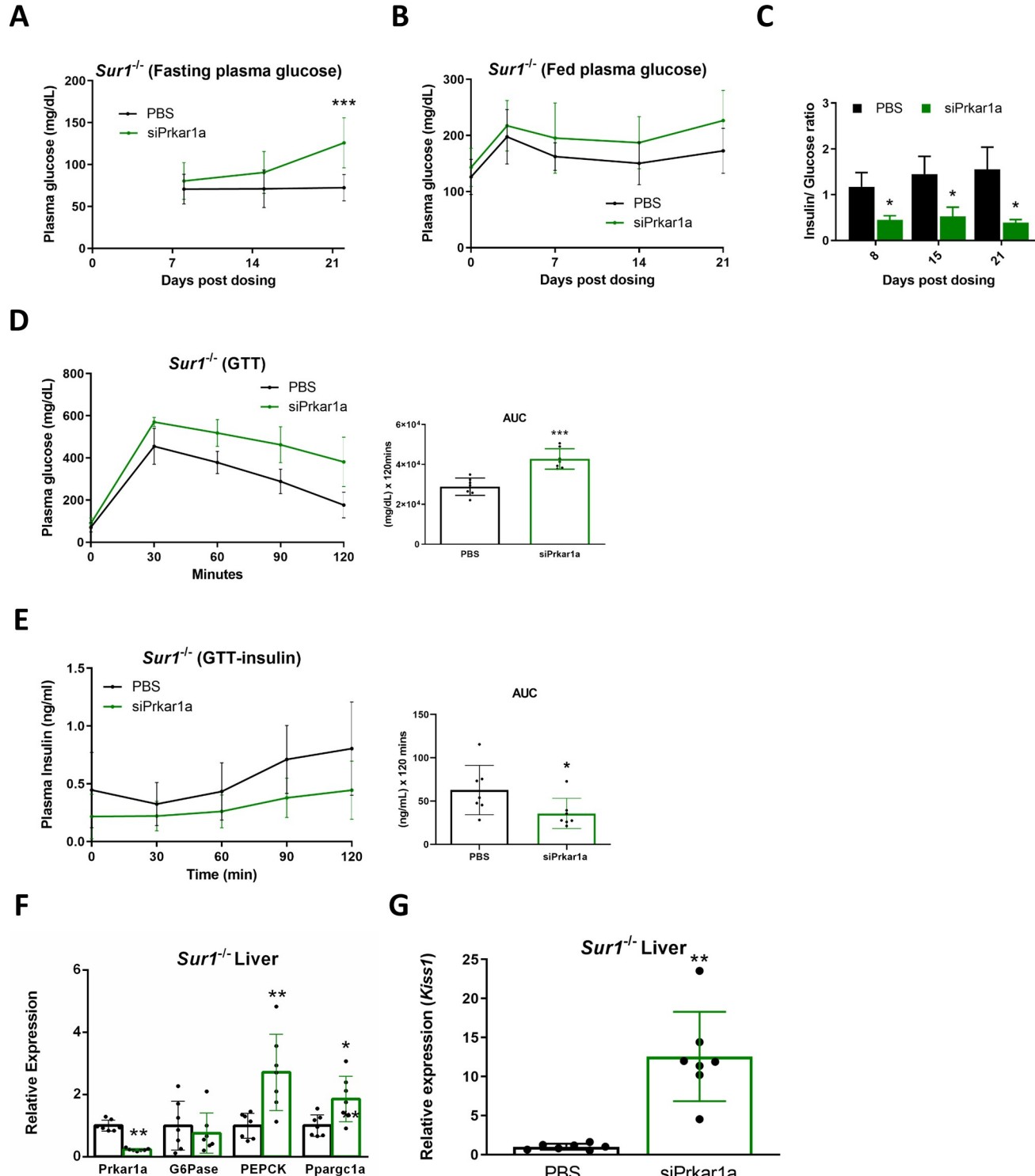

**Fig 4. Loss of Prkar1a expression attenuates fasting induced hypoglycemia in _Sur1_−/− mice.** _Sur1_−/− mice were injected subcutaneously with siPrkar1a (AD-76410, 1 mg/kg). (A) Fasting plasma glucose levels were assessed after a 16 hour overnight fast at 8, 15, and 22 days post dosing. (B) Fed plasma glucose levels were measured with ad libitum feeding at 0, 3, 7, 14, and 21 days post dosing. (C) Insulin/ glucose ratios were calculated at day 8, 15, and 21 days post siRNA injection. (D) Glucose tolerance test (GTT) was administered to siPrkar1a (AD-76410, 1 mg/kg) injected mice and controls 21 days post dosing after a 16 hour overnight fast. Area under the curve (AUC) was calculated. (E) Blood samples were collected during GTT time points to assess plasma insulin levels after a 16 hour overnight fast. Liver extracts were collected from _Sur1_−/− mice injected with siPrkar1a (AD-76410, 1 mg/kg) (green bars) or vehicle control (black bars) and assessed by qPCR for expression of (F) Prkar1a, G6pase, PEPCK (PCK), Ppargc1a or (G) Kiss1. (n = 7) Data represent mean +/- SEM. *, p ≤ 0.05; **, p ≤ 0.01; ***, p ≤ 0.001 compared to PBS control.

Examination of the $Sur1^{-/-}$ hyperinsulinism mouse model in the context of Prkar1a reduction, identified a distinct increase in not only fasting plasma glucose levels, but also a significant decrease in fasting insulin/glucose levels as well (Fig 4). Interestingly, the decrease in insulin concentration in relationship to glucose concentration indicates that the increase in plasma glucose is not solely the result of enhancement of glycogenolysis and gluconeogenesis but may also involve β-cell effects. The latter observation can be explained by the marked upregulation of Kiss1 expression in the liver in both the $Sur1^{-/-}$ mouse model and the insulin administration model. The liver has previously been shown to have important roles in glucose homeostasis through regulation of glycogen storage, gluconeogenesis, and most interestingly by suppression of insulin secretion in the pancreatic β-cell through glucagon stimulated production of the hepatokine KISS1 [14]. Of note, there have been contradictory reports demonstrating stimulatory effects of kisspeptin on insulin secretion. Assessment of the published work reveals that studies using a physiologic nanomolar concentration of kisspeptin demonstrated an inhibitory effect on insulin secretion [14, 24, 25]. The studies that showed a stimulatory effect on insulin secretion used a supraphysiological dose in the micromolar range (generally 1uM) [26–29]. Further, studies using KISS1R$^{-/-}$ mice have demonstrated that stimulation with kisspeptin beyond physiological nanomolar concentrations can occur in a KISS1R-independent mechanism [14, 30]. These results indicate potential off-target effects of kisspeptin with use of supraphysiological concentrations.

Measurement of circulating kisspeptin levels in mice and humans has been notably unreliable [31]. Unfortunately, we were unable to find reliable commercial sources for circulating kisspeptin measurements and did not investigate this mechanism further in the current study. However, we did observe decreased circulating endogenous insulin in mice dosed with Prkar1a siRNA (Figs 3B and 4E) supporting the involvement of liver-β-cell crosstalk via Kisspeptin. Glucagon deficiency and blunting of the glucagon counterregulatory response has been observed in both patients with HI and in mouse models of HI [32, 33]. The inhibitory effect of hyperinsulinism on glucagon is two-fold: mutation of the $K_{ATP}$ channel affects alpha cell membrane potential/ glucagon secretion and high circulating insulin levels directly inhibiting secretion by the alpha-cell [33, 34]. Our results indicate that it is possible to overcome these glucagon inhibitory signals and to alleviate hyperinsulinemic hypoglycemia by amplifying glucagon signaling downstream of the initiating glucagon receptor binding. Direct modification of Prkar1a expression is likely not the best therapeutic option for development in patients as it can result in the formation of myxomas in internal organs due to Carney complex complications [35]. Importantly however, our results indicate that harnessing the mechanisms of amplifying glucagon signaling through the hepatic cAMP-PKA-CREB signaling nexus is a viable option with a plethora of potential for developing therapies for HI associated hypoglycemia.

## Supporting information

**S1 Fig. Western blot analysis of phospho-PKA substrates in liver extracts from siPrka1a injected WT mice as compared to vehicle injected controls.** WT mice were injected with siRNA (AD-76410, 1mg/kg) directed against Prkar1a or PBS control every 2 weeks until liver tissue was collected 28 days post-injection of initial dose. Western blot analysis of phospho-PKA substrates was completed on the liver extracts (n = 6).
(TIF)

**S2 Fig. Reduction of Prkar1a results in hyperglycemia.** (A) WT mice were injected with siR-NAs directed against Prkar1a once at day 0 at the denoted mg/kg dose and plasma glucose levels were assessed at 0, 3, 7, 10, 14, 21, and 28 days post dosing. (B) Mice were injected with

siRNAs directed against Prkar1a every 2 weeks at the denoted mg/kg dose and plasma β hydroxyl butyrate levels were assessed at 0, 10, and 28 days post dosing. (C) Mice injected with siPrkar1a every 2 weeks (Q2W) were weighed at 0, 10, 14, 21, and 28 days post dosing. (n = 6 mice/group) Data represent mean +/- SEM.
(TIF)

**S3 Fig. siPrkar1a injection in *Sur1*<sup>-/-</sup> mice.** (A) In *Sur1*<sup>-/-</sup> mice, fasting plasma insulin levels were measured after a 16 hour overnight fast at 8, 15, and 22 days post dosing. (B) Fasting plasma β hydroxyl butyrate levels were measured after a 16 hour overnight fast at 8 and 22 days post dosing. (n = 7) Data represent mean +/- SEM.
(TIF)

**S1 Raw images.**
(TIF)

## Acknowledgments

We would like to thank Tuyen Nguyen for siRNA screening, Alnylam siRNA synthesis cores for providing siRNAs used in this study, Wendell Davis and Alnylam histology group for glycogen staining and Alnylam vivarium personnel for animal husbandry.

## Author Contributions

**Conceptualization:** Mangala M. Soundarapandian, Patrick A. Haslett, Kevin Fitzgerald, Diva D. De León.

**Data curation:** Mangala M. Soundarapandian, Christine A. Juliana.

**Formal analysis:** Mangala M. Soundarapandian, Christine A. Juliana.

**Investigation:** Mangala M. Soundarapandian, Jinghua Chai.

**Methodology:** Mangala M. Soundarapandian, Diva D. De León.

**Project administration:** Diva D. De León.

**Supervision:** Diva D. De León.

**Writing – original draft:** Christine A. Juliana.

**Writing – review & editing:** Mangala M. Soundarapandian, Patrick A. Haslett, Kevin Fitzgerald, Diva D. De León.

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
