## [Decision Letter · Decision Letter 0]

13 May 2020

PONE-D-20-09701

Amplified glucagon signaling mitigates congenital hyperinsulinism associated hypoglycemia in the Sur1-/- mouse model

PLOS ONE

Dear Dr. De Léon,

Thank you for submitting your manuscript to PLOS ONE. After careful consideration, we feel that it has merit but does not fully meet PLOS ONE’s publication criteria as it currently stands. Therefore, we invite you to submit a revised version of the manuscript that addresses the points raised during the review process.

We would appreciate receiving your revised manuscript by Jun 27 2020 11:59PM. To enhance the reproducibility of your results, we recommend that if applicable you deposit your laboratory protocols in protocols.io, where a protocol can be assigned its own identifier (DOI) such that it can be cited independently in the future. For instructions see: http://journals.plos.org/plosone/s/submission-guidelines#loc-laboratory-protocols

We look forward to receiving your revised manuscript.

Kind regards,

Michael Bader

Academic Editor

PLOS ONE

Journal Requirements:

2. To comply with PLOS ONE submissions requirements, in your Methods section, please provide additional information on the animal research and ensure you have included details on (a) how many total animals were used in the study (b) how often animal welfare checks were done, (c) efforts to alleviate suffering, and (d) the source of the Sur1 -/- mice."

4. In your methods section, please provide the catalog numbers of all antibodies used in your study. In addition, please provide a full description and sequence of the Prkar1a siRNA."

5. Our journal requires that methods are described in enough detail to allow suitably skilled investigators to fully replicate your study. Please provide a more detailed description of your glycogen staining and siRNA injection methods. If materials, methods, and protocols are well established, authors may cite articles where those protocols are described in detail, but the submission should include sufficient information to be understood independent of these references. Please revise your manuscript so that protocols are sufficiently described. For more information please see https://journals.plos.org/plosone/s/submission-guidelines#loc-materials-and-methods.

7. Thank you for stating the following in the Competing Interests section:

"MMS, PAH and KF work for Alnylam Pharamaceuticals; MMS, PAH and KF hold

shares in Alnylam Pharmaceuticals stock; DDDL received funding from Alnylam

Pharmaceuticals to conduct this study."

Reviewers' comments:

Reviewer's Responses to Questions

**Comments to the Author**

1. Is the manuscript technically sound, and do the data support the conclusions?

Reviewer #1: Yes

Reviewer #2: No

2. Has the statistical analysis been performed appropriately and rigorously? 

Reviewer #1: Yes

Reviewer #2: Yes

3. Have the authors made all data underlying the findings in their manuscript fully available?

Reviewer #1: No

Reviewer #2: Yes

4. Is the manuscript presented in an intelligible fashion and written in standard English?

Reviewer #1: Yes

Reviewer #2: Yes

5. Review Comments to the Author

Reviewer #1: The authors describe the effects of or prkar1a knockdown in the liver in WT C57Bl/6 mice and in SUR-/- mice - a model for neonatal hyperinsulinemic hypoglycemia.

Overall the manuscript has a logical structure and flow, the methods are sound and the data and adequately performed statistical analysis support in large parts the interpretation and conclusions by the authors.

Caveats to the manuscript:

The title indicates that glucagon signaling is amplified in the liver. This is inaccurate because glucagon activates multiple signaling pathways in the liver. The studies have up-regulated the PKA arm of glucagon signaling and not glucagon recptro signaling in the entirety of all possible signaling pathways. This should be corrected in the title.

The method of generating liver targeted siRNA directed to the liver is not provided. It would be difficult to assess whether these methods are adequate and rigorous.

The connection between kisspeptin and insulin secretion is mentioned but the authors do not provide sufficient data to support any connection in their experimental system. This should be clarified in the discussion section.

A time course in the development of increased in kisspeptin production and reduced insulin secretion in SUR -/- would be helpful. Such data was not generated and it may be useful for the authors to discuss this relationship

Figure legends should specify whether the data relates to studies in WT mice or to studies in SUR -/- mice.

Figure S1 legend is insufficient in describing the presented data.

Reviewer #2: Soundarapandian et al have addressed the clinical problem of hyperinsulinism-induced hypoglycemia (HI). This disorder is poorly managed and is currently treated with therapies that include replacement of glucagon and drugs which suppress insulin release. In this study, the authors propose a new approach which involves enhancing hepatic endogenous glucose production (EGP) by targeting the suppression of the regulatory subunit of PKA, Prkar1a. This molecule is a negative regulator of PKA and by disinhibition through siRNA therapeutics they propose to enhance EGP independently of glucagon administration. This is novel and intriguing.

The paper appears to have been put together in haste; some panels are jumbled in Figs 3 and 4, several supplementary figs have been added which could be consolidated, aspects of the supporting literature have been omitted and the methods are incomplete. More worrying is that many of the datasets have potentially valuable trends but fail to reach statistical validity. Thus, despite some promising actions of siPrkar1a on control rodents, the effects in HI mice are unconvincing under most conditions and fail to reach statistical significance despite adequate replicates.

1. Methods/ siPrkar1Controls. (a) There appear to be no publications relating to the (proprietary?) siRNA probesets and the authors have included no information on the sequences of their probes or cited supporting data. This needs to be included. Three probe sets are included AD-76409, AD-76410 and AD-76411 but only AD-76410 (L130) is described as liver-targeted. Please clarify. In some experiments you state which probe was used in the legend, but in others you do not – please clarify throughout. Also, the concentration of probe(s) has not been reported for a number of studies. (b) As Prkar1a is not solely associated with PKA, include additional controls to show that PKA is the only modified protein and not for example AKAPs, ARDGEFs, etc. (c) There is no controls data illustrating that scrambled probesets are ineffective. In the absence of any publications on AD-76409, AD-76410 and AD-76411, please include. (d) Are the actions of the siPrkar1a manipulations reversible. Please include. (e) I am concerned that some of the most dramatic actions – and the statistically significant effects, of siPrkar1a are only seen at the end of the study periods. How confident can we be that hepatic function has not been compromised at this point in time? (f) L135-L136. I disagree, siPrkar1treatment has a positive impact on PKA protein expression – this is not significant, but it cannot be described as ‘no discernible effect’. (g) L156-157 The actions of siPrkar1treatment on glucose levels in the PPT (Fig 1E) are not significant, this needs to me made clear in this sentence. (h) L176-L177 the statement that “AD76409 and AD-76410 led to a significant upregulation of G6Pase, PEPCK, and PGC-1α” is simply not true; AD-76410 had no action on G6Pase.

2. Figures 1D, 2E and 3D shown inconsistent actions of the siRNAs on the expression of the gluconeogenesis targets. This is best exemplified by the data involving AD-76410 and G6Pase expression. This is worrying, please clarify. The profile of targets studied is also different when the investigators used Sur1-/- mice (Fig 4F) for which there is no explanation. Please clarify and make the profiles consistent.

3. siPrkar1treatment appears to induce a consistent increase in Kiss1 gene expression and this is used by the authors to support their hypothesis that siPrkar1has the dual action of enhancing EGP and inhibiting insulin release. However, the is no discussion or citation to the fact that there is a considerable body of literature indicating that kisspeptin has a stimulatory and not inhibitory action on insulin secretion. Can the authors please clarify why this literature is missing from their paper?

4. To support your hypothesis and the data presented in the paper, please assay for kisspeptin in the pre-clinical models.

5. The data modelling HI by exogenous hyperinsulinism has weaknesses and does not fully support the authors. Figure 3A clearly illustrates that whilst siPrkar1treatment enhances plasma glucose, it cannot reverse insulin-induced hypoglycaemia. I agree that there is a trend to abrogate the actions of exogenous insulin, but this is only a trend and not significant. Why does exogenous insulin fail to elevate �-hydroxy butyrate in the control animals (Fig 3B not 3C)? It seems to work OK in the siPrkar1-treated group, but not the controls. Insulin measurements (Fig 3C not 3B) reveal a considerable range of basal (fed?) insulin levels in the mice for which there is no explanation and it is not clear whether the glucose dataset was obtained from fed or fasting mice (Fig 3A)? I also found the insulin measurement dataset confusing; the authors used different ELISA kits to detect human and mouse insulin, but it is not clear which kit has been used for the date in Fig 3C. It appears to me the assay is not able to distinguish the insulins. This needs to be made clear as both the controls and the insulin-pump animals have the same plasma insulin levels and this negates the model which should after all be hyperinsulinemic.

6. The HI Sur1-/- mice datasets detract from the findings. First, the plasma insulin levels are generally lower and not higher than the controls, which is surprising considering these mice are a model of hyperinsulinism. Second, insulin secretion was not suppressed by siPrkar1treatment which goes against their own work (Fig 3) and their explanation for how kisspeptin is relevant to the study. I agree that there is an increase in glucose levels in the mice and that this would be of benefit in a translational capacity. However, this is underpowered as the variance is high and it cannot to linked to an action on the beta-cells in this model of HI. Third, on L244-246 the authors make no comment upon the fact that siPrkar1treatment has either has no action or decreases in the expression of G6Pase which is markedly different to the what happens in non HI mice. Fourth, despite sufficient replications of data, hardly any of the in vivo profiling studies reach statistical significance or validity and this weakens rather than strengthens their arguments. Finally, without including the WT controls, it is hard to conclude the siRNA treatment reversed the hyperinsulinism in these animals. Sur1KO are not hyperinsulinemic but have lost first phase glucose-stimulated insulin secretion. Unlike humans in the fasting state, they are not hyperinsulinemic. Thus, the question whether a Prkar1a knockdown could reverse some effects of hyperinsulinism cannot be addressed in this mouse model. Thus, the authors show impact on glucose production, but not that it has an impact in face of high insulin.

6. PLOS authors have the option to publish the peer review history of their article (what does this mean?). If published, this will include your full peer review and any attached files.

Reviewer #1: No

Reviewer #2: No

---

## [Author Response · Author response to Decision Letter 0]

30 Jun 2020

Response to Reviewers:

Journal Requirements:

We have addressed the PLOS ONE style requirements and edited the manuscript (Title page, titles in sentence format, etc.) and files (dpi, names) to comply.

2. To comply with PLOS ONE submissions requirements, in your Methods section, please provide additional information on the animal research and ensure you have included details on (a) how many total animals were used in the study (b) how often animal welfare checks were done, (c) efforts to alleviate suffering, and (d) the source of the Sur1 -/- mice."

We have added the requested information to the “Materials and methods” section. 

The generation and genotyping of Sur1-/- mice were previously described [1]. Sur1-/- mice are bred and maintained in our mouse colony on a C57Bl/6 genetic background for experiments. 7 male mice, eight to ten weeks old, Sur1-/- mice were used in each group in all experiments. Mice were maintained on a 12:12-h light-dark cycle and were fed a standard rodent chow diet with free access to food and water. Animal welfare checks were completed once per day. No adverse effects to these experiments were observed or required analgesia.

3. In your methods section, please provide the catalog numbers of all antibodies used in your study. In addition, please provide a full description and sequence of the Prkar1a siRNA."

The requested antibody catalog numbers and full description/ sequence of the Prkar1a siRNA have been added to the “Materials and methods” section.

The following antibodies were used: Prkar1a (BD biosciences, Catalog #610609, 1:500), PKAC (BD Biosciences, Catalog #610980, 1:2000), ß-actin (Abcam, Catalog #ab8227, 1:2000), PKA substrate (Cell Signaling, Catalog #9624, 1:1000), Fluorescence conjugated secondary antibodies (Licor, Goat anti-rabbit, Catalog #926-32211 and Donkey anti-mouse, Catalog #926-680721:5000).

The siRNA targeting Prkar1a was designed, synthesized, and liver targeted by Alnylam Pharmaceuticals, as previously described [2, 3]. The siRNA was designed to target mouse Prkar1a mRNA, NM_021880. 

AD-76409 targets position 865-885, 5’-GAUGUAUGAAGAAUUCCUUAGUA-3’

AD-76410 targets position 873-893, 5’-AAGAAUUCCUUAGUAAAGUGUCU-3’

AD-76411 targets position 1394-1414, 5’-AAAAGUUGCUUUAUUGCACCAUU-3’

4. Our journal requires that methods are described in enough detail to allow suitably skilled investigators to fully replicate your study. Please provide a more detailed description of your glycogen staining and siRNA injection methods. If materials, methods, and protocols are well established, authors may cite articles where those protocols are described in detail, but the submission should include sufficient information to be understood independent of these references. Please revise your manuscript so that protocols are sufficiently described. For more information please see https://journals.plos.org/plosone/s/submission-guidelines#loc-materials-and-methods.

We have provided more detail for the glycogen staining and siRNA injection methods to the “Materials and methods” section.

For glycogen staining the mice were subcutaneously injected with vehicle control (1X PBS) or indicated doses of Prkar1a siRNA. Glycogen was detected in liver sections following a standardized periodic acid Schiff (PAS) staining technique. Briefly, livers fixed in 10% neutral buffered formalin were embedded in paraffin blocks. 4-micron sections were collected on glass slides, de-paraffinized, incubated with 0.5% periodic acid for 7 min, rinsed in water, and placed in Schiff’s reagent for 15 min. Finally, slides were washed with water and nuclei were stained with Modified Mayer’s Hematoxylin. 1% Diastase was used to verify that staining was specific for glycogen. All reagents were obtained from Rowley Biochemical.

For Prkar1a knockdown studies the mice were subcutaneously injected with vehicle control (1X PBS) or 1mg/kg Prkar1a siRNA every 2 weeks unless otherwise indicated. The endpoints were assessed from serum or liver tissue 28 days post dosing unless otherwise indicated.

The siRNA targeting Prkar1a was designed, synthesized, and liver targeted by Alnylam Pharmaceuticals, as previously described [2, 3].

5. PLOS ONE now requires that authors provide the original uncropped and unadjusted images underlying all blot or gel results reported in a submission’s figures or Supporting Information files. This policy and the journal’s other requirements for blot/gel reporting and figure preparation are described in detail at https://journals.plos.org/plosone/s/figures#loc-blot-and-gel-reporting-requirements and https://journals.plos.org/plosone/s/figures#loc-preparing-figures-from-image-files. When you submit your revised manuscript, please ensure that your figures adhere fully to these guidelines and provide the original underlying images for all blot or gel data reported in your submission. 

We have provided the original uncropped and unadjusted blot images in the Supplemental figures.

6. Thank you for stating the following in the Competing Interests section:

"MMS, PAH and KF work for Alnylam Pharmaceuticals; MMS, PAH and KF hold

shares in Alnylam Pharmaceuticals stock; DDDL received funding from Alnylam

Pharmaceuticals to conduct this study."

Please confirm that this does not alter your adherence to all PLOS ONE policies on sharing data and materials, by including the following statement: "This does not alter our adherence to PLOS ONE policies on sharing data and materials.”

We confirm that this does not alter our adherence to the PLOS ONE policies on sharing data and materials and have updated our Competing Interests statement.

Reviewer Comments:

Reviewer #1:

1. The title indicates that glucagon signaling is amplified in the liver. This is inaccurate because glucagon activates multiple signaling pathways in the liver. The studies have up-regulated the PKA arm of glucagon signaling and not glucagon receptor signaling in the entirety of all possible signaling pathways. This should be corrected in the title.

We have altered the title to more specifically denote the PKA arm as the active pathway:

Activation of Protein Kinase A (PKA) signaling mitigates congenital hyperinsulinism associated hypoglycemia in the Sur1-/- mouse model

2. The method of generating liver targeted siRNA directed to the liver is not provided. It would be difficult to assess whether these methods are adequate and rigorous.

We have added references to the methods we used to target the liver in the “Materials and methods” section.

To target the siRNAs to the liver, we used siRNAs conjugated to triantennary N ‐acetylgalactosamine (GalNAc) to induce robust RNAi‐mediated gene silencing in the liver, owing to uptake mediated by the asialoglycoprotein receptor (ASGPR). These methods were published previously [2, 3].

3. The connection between kisspeptin and insulin secretion is mentioned but the authors do not provide sufficient data to support any connection in their experimental system. This should be clarified in the discussion section. 

We have added further text to the “Discussion” section.

4. A time course in the development of increased in kisspeptin production and reduced insulin secretion in SUR -/- would be helpful. Such data was not generated and it may be useful for the authors to discuss this relationship.

Measurement of kisspeptin circulating levels in mice through commercially available assays has been notably unreliable due to large variations in the assay methods, ranges of detection, and lack of clarity about which isoforms of kisspeptin (i.e., KP10, KP13, KP15, KP54) are detected [4].

While we agree, we have unfortunately been unable to identify an ELISA kit that would allow for accurate assessment of the plasma KISSPEPTIN levels. The kit used in Song, et al [5] is no longer available for purchase and several other tested kits have not demonstrated good accuracy. 

5. Figure legends should specify whether the data relates to studies in WT mice or to studies in SUR -/- mice.

We have edited the figure legends to specify the mice used for the studies.

Figure 1, 2, 3 and Supplemental Figures 1 and 2 are studies completed in WT mice.

Figure 4 and Supplemental Figure 3 are studies completed in Sur1-/- mice.

6. Figure S1 legend is insufficient in describing the presented data.

We have added more details to describe the western blot presented in the Figure S1 figure legend.

WT mice were injected with siRNA (AD-76410, 1mg/kg) directed against Prkar1a or PBS control every 2 weeks until liver tissue was collected 28 days post-injection of initial dose. Western blot analysis of phospho-PKA substrates was completed on the liver extracts (n=6).

Reviewer #2:

1. Methods/ siPrkar1Controls. (a) There appear to be no publications relating to the (proprietary?) siRNA probesets and the authors have included no information on the sequences of their probes or cited supporting data. This needs to be included. Three probe sets are included AD-76409, AD-76410 and AD-76411 but only AD-76410 (L130) is described as liver-targeted. Please clarify. In some experiments you state which probe was used in the legend, but in others you do not – please clarify throughout. Also, the concentration of probe(s) has not been reported for a number of studies. (b) As Prkar1a is not solely associated with PKA, include additional controls to show that PKA is the only modified protein and not for example AKAPs, ARDGEFs, etc. (c) There is no controls data illustrating that scrambled probesets are ineffective. In the absence of any publications on AD-76409, AD-76410 and AD-76411, please include. (d) Are the actions of the siPrkar1a manipulations reversible. Please include. (e) I am concerned that some of the most dramatic actions – and the statistically significant effects, of siPrkar1a are only seen at the end of the study periods. How confident can we be that hepatic function has not been compromised at this point in time? (f) L135-L136. I disagree, siPrkar1treatment has a positive impact on PKA protein expression – this is not significant, but it cannot be described as ‘no discernible effect’. (g) L156-157 The actions of siPrkar1treatment on glucose levels in the PPT (Fig 1E) are not significant, this needs to me made clear in this sentence. (h) L176-L177 the statement that “AD76409 and AD-76410 led to a significant upregulation of G6Pase, PEPCK, and PGC-1α” is simply not true; AD-76410 had no action on G6Pase.

(a) We have added the siRNA sequences and liver targeting references to the “Materials and methods” section.

We have edited the figure legends to more clearly identify the siRNA and concentration used.

The siRNAs targeting Prkar1a (AD-76409, AD-76410, and AD-76411) were designed, synthesized, and liver targeted by Alnylam Pharmaceuticals, via conjugation to triantennary N ‐acetylgalactosamine (GalNAc) to induce robust RNAi‐mediated gene silencing in the liver, owing to uptake mediated by the asialoglycoprotein receptor (ASGPR)as previously described [2, 3]. AD-76410 (1mg/kg) was used for experiments unless otherwise noted. 

The siRNA was designed to target mouse Prkar1a mRNA, NM_021880. 

AD-76409 targets position 865-885, 5’-GAUGUAUGAAGAAUUCCUUAGUA-3’

AD-76410 targets position 873-893, 5’-AAGAAUUCCUUAGUAAAGUGUCU-3’

AD-76411 targets position 1394-1414, 5’-AAAAGUUGCUUUAUUGCACCAUU-3’

(b) In this manuscript, we sought to address whether amplifying signaling downstream of the glucagon receptor in the liver could counteract the effects of hyperinsulinism. Our goal was to activate PKA which is a well-established and central player in the glucagon pathway and activates several key downstream mediators of glucagon, including kisspeptin expression. Guanine nucleotide exchange factors (GEFs) would be associated with cAMP upstream of PRKAR1a inhibition of PKA. The role of A-kinase anchoring proteins (AKAPs) are associated with anchoring PKA to confer compartmentalization of PKA activation. While these proteins have interesting roles in the PKA pathway, investigation of the multitude of proteins identified in the PKA pathway is beyond the scope of our manuscript and is not directly relevant to the hypothesis.

(c) While we did not include a scrambled control probe set, probe AD-76411 serves as a functional negative control due to its inability to decrease the Prkar1a mRNA or plasma glucose levels significantly.

(d) Unfortunately, we do not know if the actions of the siPrkar1a are reversible. Treated mice were euthanized for tissue harvest at the end of the experiment.

(e) The length of time needed to see effects of the siPrkar1a effects are due to the time necessary for significant downstream effects to occur (i.e. changes in transcriptional and protein expression). We are highly confident that hepatic function is not compromised in these experiments. Conditions that cause insulin related hepatic dysfunction such as metabolic syndrome and insulin resistance require years to compromise liver function. The short duration of this experiment (several weeks) is not enough time to develop hepatic dysfunction.

(f) We agree that “no discernible effect” was not the right way to describe the data. We have edited the text to instead say “while not having a significant effect on catalytic (PKAc) protein levels.

(g) We have added the following underlined words to clarify:

Functionally, the loss of Prkar1a resulted in an increase in glycogenolysis and gluconeogenesis as observed by a significant reduction in liver glycogen (Fig 1D) and though not statistically significant, a trend of increased conversion of the gluconeogenic precursor pyruvate to glucose in a pyruvate tolerance test (Fig 1E) was observed in mice injected with siPrkar1a.

(h) We have edited the text to remedy this mistake.

The most potent siRNAs against Prkar1a (AD-76409 and AD-76410) led to a significant upregulation of gluconeogenesis genes G6Pase, PEPCK, and PGC-1α (Fig 2E). AD-76409 also significantly increased G6Pase expression.

2. Figures 1D, 2E and 3D shown inconsistent actions of the siRNAs on the expression of the gluconeogenesis targets. This is best exemplified by the data involving AD-76410 and G6Pase expression. This is worrying, please clarify. The profile of targets studied is also different when the investigators used Sur1-/- mice (Fig 4F) for which there is no explanation. Please clarify and make the profiles consistent.

In both figures (Figure 1D and 2E) G6Pase shows upregulation, though admittedly, the average is less and not significant in Figure 2E (AD-76410). Variability in transcript levels and a smaller sample size is the likely explanation for the lack of significance with AD-76410. However, it is important to take into consideration that the upward direction of regulation, even if not significant, and the consistency of the upregulation of the other two markers (Ppargc1a and PEPCK) consistently support our hypothesis.

From our data, it appears that siPrkar1a is not able to overcome the suppression of G6pase in the setting of increased insulin (endogenous or exogenous). Importantly though, the upregulation of the other two gluconeogenesis markers remain consistent in the Sur1-/- mouse model. 

The profile of targets studied is the same for Figure 1C, 2E, and 4F (G6pase, PEPCK, and Ppargc1a). Inadvertently during the figure creation process, some of the alternate names were used for the figure labels (i.e. Ppargc1a (PGC1α); PEPCK (PCK)). We have edited the figure labels for clarity and consistency. 

3. siPrkar1 treatment appears to induce a consistent increase in Kiss1 gene expression and this is used by the authors to support their hypothesis that siPrkar1has the dual action of enhancing EGP and inhibiting insulin release. However, the is no discussion or citation to the fact that there is a considerable body of literature indicating that kisspeptin has a stimulatory and not inhibitory action on insulin secretion. Can the authors please clarify why this literature is missing from their paper?

We are aware of the contradictory studies published that demonstrate opposing effects of treatment with kisspeptin with either an inhibitory or stimulatory action on insulin secretion. One potential proposed explanation was that different isoforms of kisspeptin (KP10 vs KP13) may have different effects on insulin secretion. The 154 amino acid pre-propeptide encoded by Kiss1 is proteolytically processed and results in the 54-amino acid product kisspeptin-54 (KP54, metastin), but also three C-terminal fragments, kisspeptin-14 (KP14), kisspeptin-13 (KP13), and kisspeptin-10 (KP10), which all share the same 10 amino acid amidated sequence. The smallest fragment, KP10, is sufficient to bind and activate the KISS1 receptor (KISS1R) [6]. However, a study comparing KP10 versus KP13 did not demonstrate any functional changes between the two isoforms [7].

The more likely explanation of the contradictory stimulatory/ inhibitory actions of kisspeptin on insulin secretion is the amount of kisspeptin used in the study. Plasma kisspeptin concentrations in HFD and Leprdb/db mice, which was significantly higher than controls, was 0.5 – 1 nM and 7 – 10 nM respectively [5]. Thus, the physiological circulating levels of kisspeptin in mice is measured to be in the very low nanomolar range. Assessment of the published work reveals that studies using a nanomolar concentration of kisspeptin demonstrated an inhibitory effect on insulin secretion [5, 8, 9]. The studies that showed a stimulatory effect on insulin secretion used a supraphysiological dose in the micromolar range (generally 1uM) [7, 10-12]. Further, studies using KISS1R-/- mice have demonstrated that stimulation can occur in a KISS1R-independent mechanism [5, 13]. These results indicate potential off-target effects of kisspeptin with use of supraphysiological concentrations.

We originally did not include discussion of the studies indicating stimulatory effects of kisspeptin, since they appear to be a result of off-target effects by supraphysiological concentrations and our studies involve endogenously expressed kisspeptin. However, we understand the necessity for clarification of the contradictory studies and have included it in our “Discussion” section. 

4. To support your hypothesis and the data presented in the paper, please assay for kisspeptin in the pre-clinical models.

Measurement of kisspeptin circulating levels in mice through commercially available assays has been notably unreliable due to large variations in the assay methods, ranges of detection, and lack of clarity about which isoforms of kisspeptin (i.e., KP10, KP13, KP15, KP54) are detected [4]. Unfortunately, we have been unable to identify an ELISA kit that would allow for accurate assessment of the plasma KISSPEPTIN levels in mice. We are limited by the commercially available ELISA kits. The kit used in Song, et al [2] is no longer available for purchase and several other tested kits have not demonstrated good accuracy.

5. The data modelling HI by exogenous hyperinsulinism has weaknesses and does not fully support the authors. Figure 3A clearly illustrates that whilst siPrkar1treatment enhances plasma glucose, it cannot reverse insulin-induced hypoglycaemia. I agree that there is a trend to abrogate the actions of exogenous insulin, but this is only a trend and not significant. Why does exogenous insulin fail to elevate ß-hydroxy butyrate in the control animals (Fig 3B not 3C)? It seems to work OK in the siPrkar1-treated group, but not the controls. Insulin measurements (Fig 3C not 3B) reveal a considerable range of basal (fed?) insulin levels in the mice for which there is no explanation and it is not clear whether the glucose dataset was obtained from fed or fasting mice (Fig 3A)? I also found the insulin measurement dataset confusing; the authors used different ELISA kits to detect human and mouse insulin, but it is not clear which kit has been used for the date in Fig 3C. It appears to me the assay is not able to distinguish the insulins. This needs to be made clear as both the controls and the insulin-pump animals have the same plasma insulin levels and this negates the model which should after all be hyperinsulinemic.

Insulin suppresses lipolysis and ketogenesis, thus, in insulin-mediated hypoglycemia, ß-hydroxybutyrate is suppressed. Our data indicates that during insulin-induced hypoglycemia, downregulation of Prkar1a allows ketogenesis to be activated. Controls (non-insulin treated) are not hypoglycemic, so β-hydroxy butyrate is not expected to be increased. 

Plasma insulin concentrations in Figure 3C was assessed in mice fasted for 5 hours as stated in the figure legend. While there are a few outliers, the insulin levels fall within the expected range of plasma insulin. We have corrected the figure placement (Figure 3B is Insulin levels; Figure 3C is β- hydroxybutyrate levels).

Plasma glucose levels in Figure 3A were obtained after a 5 hour fast. This information has been added to the figure legend.

The Insulin ELISA kit used for Figure 3B detects both mouse and human. The insulin can be cleared rapidly, and like other hormones, plasma insulin concentration must be interpreted in the context of the metabolic state, particularly of the plasma glucose concentration. Insulin levels in the Sur1-/- (and also in humans with congenital hyperinsulinism) are not overtly elevated, but fail to be suppressed in the presence of hypoglycemia. Although the plasma insulin concentration is not markedly different between mice treated with insulin or not, there is a clear hypoglycemic effect of the treatment as shown in Figure 3A. We gave the mice the minimum dose of insulin necessary to achieve hypoglycemia, which may not be easy to detect by ELISA. When we optimized this model, we did try higher exogenous insulin levels but the mice became so hypoglycemic that we couldn’t complete experiments. Insulin doses lower than 0.2 U do not induce hypoglycemia.

6. The HI Sur1-/- mice datasets detract from the findings. First, the plasma insulin levels are generally lower and not higher than the controls, which is surprising considering these mice are a model of hyperinsulinism. Second, insulin secretion was not suppressed by siPrkar1treatment which goes against their own work (Fig 3) and their explanation for how kisspeptin is relevant to the study. I agree that there is an increase in glucose levels in the mice and that this would be of benefit in a translational capacity. However, this is underpowered as the variance is high and it cannot to linked to an action on the beta-cells in this model of HI. Third, on L244-246 the authors make no comment upon the fact that siPrkar1treatment has either has no action or decreases in the expression of G6Pase which is markedly different to the what happens in non HI mice. Fourth, despite sufficient replications of data, hardly any of the in vivo profiling studies reach statistical significance or validity and this weakens rather than strengthens their arguments. Finally, without including the WT controls, it is hard to conclude the siRNA treatment reversed the hyperinsulinism in these animals. Sur1KO are not hyperinsulinemic but have lost first phase glucose-stimulated insulin secretion. Unlike humans in the fasting state, they are not hyperinsulinemic. Thus, the question whether a Prkar1a knockdown could reverse some effects of hyperinsulinism cannot be addressed in this mouse model. Thus, the authors show impact on glucose production, but not that it has an impact in face of high insulin

1) Insulin levels, like other hormones (such as parathyroid hormone in response to calcium) have to be interpreted in the context of the metabolic state. Insulin levels in Sur1-/- mice (and humans with congenital hyperinsulinism) are not overtly elevated, yet they are inappropriate in the setting of hypoglycemia and reflect a marked disturbance in regulated insulin secretion with serious clinical consequences [14, 15]. While the Sur1-/- mouse model’s phenotype is milder compared to the human phenotype, all cardinal features of KATP-hyperinsulinism are reproduced by this model, including fasting hypoglycemia and impaired glucose stimulated insulin secretion. Similarly, isolated islets from the Sur1-/- mouse and human KATP-hyperinsulinism islets exhibit all abnormalities expected from the lack of functional KATP channels: elevated cytosolic calcium, high baseline insulin secretion and impaired glucose stimulated insulin secretion. Thus, we believe that this model is appropriate for these proof-of-concept studies. Furthermore, we have previously used this model for proof-of-concept studies of other potential therapies [16] that were then successfully translated to clinical studies in affected individuals with KATP-hyperinsulinism [17].

2) Insulin secretion was suppressed by siPrkar1a treatment. We have added an area under the curve (AUC) for the plasma insulin levels taken during the GTT (Figure 4E) to make this clearer.

3) While siPrkar1a increases G6Pase in the absence of increased insulin (Figure 1C), in the presence of increased insulin (endogenous or exogenous) siPrkar1a is not able to overcome the suppression of G6pase in the setting of increased insulin (hyperinsulinism or exogenous).

We have added this to the manuscript to describe the effect of insulin and siPrkar1a on G6Pase expression.

4) As we have previously shown, in Sur1-/- mice the hypoglycemic phenotype is triggered by fasting. Although we did not include wild type controls on this experiment, from previous studies, fasting plasma glucose in Sur1-/- is significantly lower than wild type littermates (59.4 ± 1.5 vs 75 ± 1.8 mg/dL, p<0.0001) [16]. In this study, fasting glucose concentrations of control treated mice are similarly low and siPrkar1a significantly increased the fasting glucose 21 days injection (Figure 4A, D). We were remiss to not include the AUC for the plasma insulin levels after fasting in the GTT (Figure 4E), which shows a significant decrease in plasma insulin the Sur1-/- siPrkar1a injected mice. 

Overall, we have demonstrated a reversal of the hyperinsulinemic state of the Sur1-/- with siPrkar1a through the significant increase in plasma glucose (Figure 4A and D) as well as a significant decrease in the level of secreted insulin (Figure 4E AUC).

References

1. Shiota C, Larsson O, Shelton KD, Shiota M, Efanov AM, Hoy M, et al. Sulfonylurea receptor type 1 knock-out mice have intact feeding-stimulated insulin secretion despite marked impairment in their response to glucose. J Biol Chem. 2002;277(40):37176-83. doi: 10.1074/jbc.M206757200. PubMed PMID: 12149271.

2. Rajeev KG, Nair JK, Jayaraman M, Charisse K, Taneja N, O'Shea J, et al. Hepatocyte-specific delivery of siRNAs conjugated to novel non-nucleosidic trivalent N-acetylgalactosamine elicits robust gene silencing in vivo. Chembiochem. 2015;16(6):903-8. doi: 10.1002/cbic.201500023. PubMed PMID: 25786782.

3. Nair JK, Willoughby JL, Chan A, Charisse K, Alam MR, Wang Q, et al. Multivalent N-acetylgalactosamine-conjugated siRNA localizes in hepatocytes and elicits robust RNAi-mediated gene silencing. J Am Chem Soc. 2014;136(49):16958-61. doi: 10.1021/ja505986a. PubMed PMID: 25434769.

4. Hussain MA, Song WJ, Wolfe A. There is Kisspeptin - And Then There is Kisspeptin. Trends Endocrinol Metab. 2015;26(10):564-72. doi: 10.1016/j.tem.2015.07.008. PubMed PMID: 26412157; PubMed Central PMCID: PMCPMC4587393.

5. Song WJ, Mondal P, Wolfe A, Alonso LC, Stamateris R, Ong BW, et al. Glucagon regulates hepatic kisspeptin to impair insulin secretion. Cell Metab. 2014;19(4):667-81. doi: 10.1016/j.cmet.2014.03.005. PubMed PMID: 24703698; PubMed Central PMCID: PMCPMC4058888.

6. Kotani M, Detheux M, Vandenbogaerde A, Communi D, Vanderwinden JM, Le Poul E, et al. The metastasis suppressor gene KiSS-1 encodes kisspeptins, the natural ligands of the orphan G protein-coupled receptor GPR54. J Biol Chem. 2001;276(37):34631-6. doi: 10.1074/jbc.M104847200. PubMed PMID: 11457843.

7. Bowe JE, Foot VL, Amiel SA, Huang GC, Lamb MW, Lakey J, et al. GPR54 peptide agonists stimulate insulin secretion from murine, porcine and human islets. Islets. 2012;4(1):20-3. doi: 10.4161/isl.18261. PubMed PMID: 22192948.

8. Silvestre RA, Egido EM, Hernandez R, Marco J. Kisspeptin-13 inhibits insulin secretion without affecting glucagon or somatostatin release: study in the perfused rat pancreas. J Endocrinol. 2008;196(2):283-90. doi: 10.1677/JOE-07-0454. PubMed PMID: 18252951.

9. Vikman J, Ahren B. Inhibitory effect of kisspeptins on insulin secretion from isolated mouse islets. Diabetes Obes Metab. 2009;11 Suppl 4:197-201. doi: 10.1111/j.1463-1326.2009.01116.x. PubMed PMID: 19817802.

10. Hauge-Evans AC, Richardson CC, Milne HM, Christie MR, Persaud SJ, Jones PM. A role for kisspeptin in islet function. Diabetologia. 2006;49(9):2131-5. doi: 10.1007/s00125-006-0343-z. PubMed PMID: 16826407.

11. Bowe JE, King AJ, Kinsey-Jones JS, Foot VL, Li XF, O'Byrne KT, et al. Kisspeptin stimulation of insulin secretion: mechanisms of action in mouse islets and rats. Diabetologia. 2009;52(5):855-62. doi: 10.1007/s00125-009-1283-1. PubMed PMID: 19221709.

12. Schwetz TA, Reissaus CA, Piston DW. Differential stimulation of insulin secretion by GLP-1 and Kisspeptin-10. PLoS One. 2014;9(11):e113020. doi: 10.1371/journal.pone.0113020. PubMed PMID: 25401335; PubMed Central PMCID: PMCPMC4234631.

13. Liu X, Herbison A. Kisspeptin regulation of arcuate neuron excitability in kisspeptin receptor knockout mice. Endocrinology. 2015;156(5):1815-27. doi: 10.1210/en.2014-1845. PubMed PMID: 25756309.

14. Palladino AA, Bennett MJ, Stanley CA. Hyperinsulinism in infancy and childhood: when an insulin level is not always enough. Clin Chem. 2008;54(2):256-63. Epub 2007/12/25. doi: 10.1373/clinchem.2007.098988. PubMed PMID: 18156285.

15. De Leon DD, Stanley CA. Determination of insulin for the diagnosis of hyperinsulinemic hypoglycemia. Best Pract Res Clin Endocrinol Metab. 2013;27(6):763-9. Epub 2013/11/28. doi: 10.1016/j.beem.2013.06.005. PubMed PMID: 24275188; PubMed Central PMCID: PMCPMC4141553.

16. De Leon DD, Li C, Delson MI, Matschinsky FM, Stanley CA, Stoffers DA. Exendin-(9-39) corrects fasting hypoglycemia in SUR-1-/- mice by lowering cAMP in pancreatic beta-cells and inhibiting insulin secretion. J Biol Chem. 2008;283(38):25786-93. Epub 2008/07/19. doi: 10.1074/jbc.M804372200. PubMed PMID: 18635551; PubMed Central PMCID: PMCPMC3258866.

17. Calabria AC, Li C, Gallagher PR, Stanley CA, De Leon DD. GLP-1 receptor antagonist exendin-(9-39) elevates fasting blood glucose levels in congenital hyperinsulinism owing to inactivating mutations in the ATP-sensitive K+ channel. Diabetes. 2012;61(10):2585-91. Epub 2012/08/03. doi: 10.2337/db12-0166. PubMed PMID: 22855730; PubMed Central PMCID: PMCPMC3447900.

---

## [Decision Letter · Decision Letter 1]

16 Jul 2020

Activation of Protein Kinase A (PKA) signaling mitigates congenital hyperinsulinism associated hypoglycemia in the Sur1-/- mouse model

PONE-D-20-09701R1

Dear Dr. De Léon,

We’re pleased to inform you that your manuscript has been judged scientifically suitable for publication and will be formally accepted for publication once it meets all outstanding technical requirements.

Kind regards,

Michael Bader

Academic Editor

PLOS ONE

Additional Editor Comments (optional):

Reviewers' comments:

Reviewer's Responses to Questions

**Comments to the Author**

1. If the authors have adequately addressed your comments raised in a previous round of review and you feel that this manuscript is now acceptable for publication, you may indicate that here to bypass the “Comments to the Author” section, enter your conflict of interest statement in the “Confidential to Editor” section, and submit your "Accept" recommendation.

Reviewer #1: All comments have been addressed

2. Is the manuscript technically sound, and do the data support the conclusions?

Reviewer #1: Yes

3. Has the statistical analysis been performed appropriately and rigorously? 

Reviewer #1: Yes

4. Have the authors made all data underlying the findings in their manuscript fully available?

Reviewer #1: Yes

5. Is the manuscript presented in an intelligible fashion and written in standard English?

Reviewer #1: Yes

6. Review Comments to the Author

Reviewer #1: The authors have addressed the comments by this reviewer

The short title is misleading. It would be more appropriate to indicate that Prkar1a KD mitigates HI

The lack of availability of Kisspeptin ELISA is problematic, but the authors can not be held to a standard that is impossible to meet.

7. PLOS authors have the option to publish the peer review history of their article (what does this mean?). If published, this will include your full peer review and any attached files.

Reviewer #1: No

---

## [Editor Report · Acceptance letter]

20 Jul 2020

PONE-D-20-09701R1 

Activation of Protein Kinase A (PKA) signaling mitigates congenital hyperinsulinism associated hypoglycemia in the Sur1^-/-^ mouse model 

Dear Dr. De León:

I'm pleased to inform you that your manuscript has been deemed suitable for publication in PLOS ONE. Congratulations! Your manuscript is now with our production department. 

Kind regards, 

on behalf of

Prof. Michael Bader 

Academic Editor

PLOS ONE